# On the Negative Impact of Mycorrhiza Application on Maize Plants (*Zea mays*) Amended with Mineral and Organic Fertilizer

**DOI:** 10.3390/microorganisms11071663

**Published:** 2023-06-26

**Authors:** Matthias Thielicke, Julian Ahlborn, Bettina Eichler-Löbermann, Frank Eulenstein

**Affiliations:** 1Department Sustainable Grassland Systems, Leibniz Center for Agricultural Landscape Research (ZALF), Gutshof 7, 14641 Paulinenaue, Germany; matthias.thielicke@zalf.de; 2Botany Division, Senckenberg Museum of Natural History Görlitz, Am Museum 1, 02806 Görlitz, Germany; 3Department of Agronomy and Crop Science, Faculty of Agricultural and Environmental Sciences, University of Rostock, 18051 Rostock, Germany

**Keywords:** *Rhizoglomus intraradices*, parasitism, *Bacillus velezensis*, microgranules, corn, biostimulants

## Abstract

Many studies describe the positive effect of mycorrhiza, but few report on negative effects. Furthermore, there is a research gap on the mechanisms under which conditions the symbiotic mycorrhizal plant interaction or a parasitic one predominates. The study was conducted as a field experiment over three years to investigate the effect of mycorrhiza (*Rhizoglomus intraradices*) and soil bacteria applications on fertile soil. A standard fertilizer (diammonium phosphate) and two microgranular fertilizers (mineral and organomineral) were applied alone or in combination with the biostimulants mycorrhiza and/or soil bacteria (*Bacillus velezensis*). The application of the mycorrhiza as the only biostimulant resulted in lower yields compared to all fertilizer variants without the mycorrhiza or with mycorrhiza in combination with soil bacteria in the dry years 2015 (*p* = 0.0241) and 2016 (*p* = 0.0003). The usage of soil bacteria alone, or soil bacteria with fertilizer, resulted in few occasional significant differences. The combination with soil bacteria raised the yield of mycorrhiza-treated fertilizer variants to a significant extent in 2015 (*p* = 0.0007) and 2016 (*p* = 0.0019). The negative effects of mycorrhiza application in this study were alleviated by the simultaneous use of soil bacteria. Treatments with organomineral microgranular fertilizer, which were expected to promote the naturally occurring soil microbiome more than the mineral fertilizer variants, were most negatively affected by the mycorrhiza. We hypothesize that the naturally occurring microbiome of the study site was already optimal for maize plants, and thus the successful introduction of other microorganisms through the application of the mycorrhiza and soil bacteria tended not to be beneficial. The present study is the first report on the negative influence of arbuscular mycorrhiza on maize yields gained with a standard fertilizer (diammonium phosphate) and two microgranular fertilizer, and the alleviation of that impact by combined application of soil bacteria. We conclude that the application of the used biostimulants may have negative impacts on maize yield if the soil is already rich in nutrients and water is the limiting factor.

## 1. Introduction

Biostimulants are products used in agriculture to raise crop resistance to pathogens or abiotic factors and to support plants’ nutrient uptake [1]. Compared to fertilizer treatments, the nutrient input from biostimulants per surface unit of measurement is negligible. Thus, biostimulants can be used to maintain or raise agricultural productivity while reducing the use of nutrients and pesticides [2,3]. Positive influences of applying mycorrhizae, a well-known representative of a biostimulant, are well described in the literature [4,5,6,7], but little information is available about negative or insignificant experimental results [8,9,10,11]. The imbalance between the numerous positive reports and the few null or negative results is a known problem in science [12,13]. To understand complex processes, such as the effect of mycorrhiza on plants, null and negative results can be as important as positive results, and should be included in specialized considerations on that topic. Otherwise, the result is the above-mentioned imbalance in scientific results, which may lead researchers or policymakers to draw the wrong conclusions [14]. Some authors conclude that knowledge about the mycorrhizosphere is sparse, and the processes in it are multifactorial, driven by both abiotic and biotic influences [15,16]. Mycorrhiza–plant interactions clearly differ between plants, but can also vary within the same plant species, and are also dependent on mycorrhiza species [17,18]. Furthermore, positive and negative interactions between mycorrhizae and soil bacteria need to be considered [19,20,21,22,23,24,25]. Additionally, abiotic factors can influence the effect of mycorrhiza on plants. Authors such as Tinker et al., 1994 [26], Ruiz-Lozano et al., 2003 [4] or Aikio and Ruotsalainen, 2002 [27] described how plants generally benefit from mycorrhizal colonization if nutrients are limited, but the water and light factors do not limit carbon acquisition for both plant growth and the supply of mycosymbiont. However, it is documented that the proportional benefit of maize plants inoculated with *Glomus etunicatum* is growing with increasing osmotic stress [28]. The application of mycorrhiza in maize cultivation appears to be especially promising due to the importance of this crop in agriculture and the high mycorrhizal development on *Zea mays* [10,29]. Several studies report positive effects of various mycorrhizal fungi on plant performance, nutrient uptake and growth of maize [30,31,32]. Recent studies document the specific interaction of the mycorrhiza species used in this study (*Glomus intraradices*) and maize. This interaction was characterized by facilitated nutrient uptake, higher physiological plant performance and the improvement in soil structure [29,33].

The market for biostimulants has been rapidly growing for more than a decade [34], but mycorrhizal products and other biostimulants are not established on farms in central Europe to the same extent. A critical assessment in relation to the growing conditions is important to reveal interrelated factors in the mycorrhizosphere, and to predict the effects of mycorrhizal products in agricultural practice or forestry. The present contribution deals with the question as to whether the influence of a mycorrhiza on maize yield gained with a standard fertilizer (diammonium phosphate) and two microgranular fertilizer may be negative. Further we hypothesize that the interaction between mycorrhiza and plants can be influenced by a combined application with soil bacteria.

## 2. Materials and Methods

### 2.1. Study Area and Experimental Setup

The experiment was carried out as field trial with 12 fertilizer variants. These consist of four different nutrient treatments with either (1) no fertilizer as a control (only pre-treatment with pig slurry), (2) the mineral fertilizer diammonium phosphate (DAP), (3) the microgranular fertilizer Wolf-nutraxP^®^ (WolfNP) (MTD Products Inc., 5903 Grafton Road, Valley City, OH, USA) or (4) the organomineral microgranular fertilizer Startec (*De Ceuster Meststoffen* NV (*DCM*), Bannerlaan 79, 2280 Grobbendonk, Belgium). The nutrient treatments 1–4 were applied singly, or with the biostimulants mycorrhiza (abbreviated as M), *Bacillus velezensis* (RhizoVital42^®^TB), (ABiTEP GmbH, Glienicker Weg 185, 12489 Berlin, Germany) (abbreviated as Bac) or in parallel with both biostimulants (abbreviated as Bac_M).

Each variant was repeated four times on plots measuring 13 m in length and 3 m in width. The plots were the same during the three years of the experiment. The field study has been conducted near Wadersloh in western Germany (51.717486, 8.240336). The region is classified as having a European Atlantic climate, as defined by Köppen and Geiger [35], characterized by mild winters and moderate summer temperatures. The average precipitation per year for Wadersloh is 673 mm, and the average annual temperature is 10.4 °C. Around 53% of the annual rainfall is during the maize vegetation period from April to September. Loamy sand with a humus content of 2.5%, an effective field capacity of 9% (volumetric) and pH of 5.7 is present on the study site. A calcium-acetate-lactate extraction (CAL-extract) of the soil revealed moderate amounts of phosphorus (6.6 mg 100 g^−1^), potassium (7.5 mg 100 g^−1^) and magnesium (3.6 mg 100 g^−1^).

The site has been used for maize cultivation for years. Regular tillage operations performed before sowing in early spring (March/April) involved plowing at a depth of 25 cm and, in two cases, grubbing to a depth of 6–10 cm, respectively. The following plant protectants were used, following the long term practice realized on the respective field: Gardo Gold^®^ (metolachlor and terbuthylazine) applied at the end of April in the form of two liters per hectare and at the beginning of June in the form of one liter combined with 0.3 L of Buctril^®^ (bromoxynil), 0.9 L of MeisTer^®^ (foramsulfuron, thiencarbazone, iodosulfuron, cyprosulfamide), 1 L of manganese nitrate and 0.2 L of Zeavit (oat-derived phytochemicals). The study site was pre-treated with pig slurry (20 m^3^ ha^−1^) containing 8.6 kg of total N, 2 kg of P, 4.9 kg of K, 2.4 kg of S and 10.8 kg of Mg per m^3^.

The maize cultivar *Farmpilot* (Agromais GmbH, Grothues 6, 48351 Everswinkel, Germany) was sown with a density of 8.5 seeds per square meter, using the AMAZONE single corn seeder system (EDX 6000-2C precision air seeder). DAP fertilizer was applied in a band 12 cm below the soil surface, in an amount of 100 kg ha^−1^. In the same amount, the microgranular fertilizer WolfNP was applied, while Startec has been used with 25 kg ha^−1^, following manufacturer’s prescription. Both microgranular fertilizers were put a few centimeters beneath the corn, using the above described single corn seeder.

DAP contains 18% total N, all in the form of NH_4_-N, and 20% P. Startec can be classified as an organomineral fertilizer, of which 80% (of the original substance) is made up of the organic industrial by-products of oil cake and de-glued bonemeal. Mineral components in Startec are ammonium phosphate, ammonium sulfate, EDTA-chelated Fe, Mn, Zn, zinc sulfate and zinc oxide. WolfNP can be classified as a mineral microgranular fertilizer, and contains ammonium sulfate, kieserite (magnesium sulfate as a monohydrate) and Nu-TraxP. The latter is a fine-grained powder that can be dissolved in water as a coating for the product. The nutrient inputs through different fertilizer treatments in the experiments are given in Table 1.

The mycorrhiza species *Rhizoglomus intraradices* has been grown in a greenhouse with maize as host on expanded clay with a bulk density of 330 kg per m^3^. The mycorrhiza-maize culture has been permanently cultivated since 1992 by Mitscherlich Academy for Soil Fertility (MITAK GmbH, Professor-Mitscherlich-Allee 1, 14641 Paulinenaue, Germany). The mycorrhiza was applied at 100 kg ha^−1^ to introduce 15.15 million spores per ha. The application rate of the soil bacteria product RhizoVital42^®^TB (ABiTEP GmbH, Glienicker Weg 185, 12489 Berlin, Germany) in the present study was 10 kg ha^−1^ (>1 bn colony-forming units of *Bacillus velezensis* per mg, Strain FZB42 (DSM No. 23117)). All the products were poured into separate chambers of an AMAZONE precision seeder for precise plot-specific application in the working step as a fertilizer treatment.

### 2.2. Data Sampling and Statistical Analysis

Manual harvesting was performed by randomly removing 20 plants per plot. The corncobs and the shoots were weighed and shredded separately, using a garden shredder (AL-KO Master 32–40). Shredded material was dried to determine the dry matter content.

To ensure that the data distribution was normal, the yield and corncob ratio were transformed using an exponential function. Differences between the fertilizer variants were tested using a pairwise Student’s *t*-test. All statistical analyses were performed in R [36]. For data selection, the package dplyr [37] was used. Visualization in R was conducted by using the package ggplot2 [38].

## 3. Results

### Interactions between Applied Mycorrhiza (M) and Bacteria (Bac) Products, and the Influence of Their Application on Different Fertilizers

When mycorrhiza was applied without bacteria, this resulted generally in lower yields compared to the collective yield gained with all fertilizer variants without mycorrhiza, or mycorrhiza in combination with soil bacteria, in 2015 (*p* = 0.0241) and 2016 (*p* < 0.001). The influence of mycorrhiza on the particular yields gained with DAP and WolfNP was neutral in 2015 and 2017. In 2016, the combination with mycorrhiza resulted in significantly lower yields on plots fertilized with DAP (*p* = 0.0301) and WolfNP (*p* = 0.0106) compared to single treatments of the two mineral fertilizer. For Startec, drops in yield, if combined with mycorrhiza, reached a significant extent in 2016 (*p* < 0.001) and 2017 (*p* = 0.0053) (Figure 1). The application of mycorrhiza on control plots did not result in differences in yield of statistical significance.

Compared to the single application of mycorrhiza (M), the combination of the latter with soil bacteria (Bac_M) raised the yield to a significant extent in 2015 (*p* < 0.001) and 2016 (*p* = 0.0019) over the collective yield gained with all fertilizer variants (Figure 2).

No significant differences in collective yield of all fertilizer variants were found in 2016 or 2017 for treatment with soil bacteria as the only biostimulant (alone and with fertilizer) compared to the combined application of soil bacteria plus mycorrhiza, while soil bacteria alone (without mycorrhiza) performed significantly worse in 2015 (*p* = 0.0287).

The combination of both mycorrhiza and soil bacteria yielded a 16.4% (2.6 t ha^−1^) higher crop in 2015 on DAP-fertilized plots (*p* = 0.0303) compared to DAP-fertilization without biostimulants, but resulted also in a lower corncob ratio (*p* < 0.001). The yield gained with WolfNP in combination with Bac_M was 10.2% lower in 2016 (*p* = 0.0066), while there were no statistically significant differences in 2015 and 2017. Startec’s yield was higher without Bac_M in 2016 and 2017. These differences, ranging from 10.1% (2.6 t) to 16.3% (3.8 t), were of statistical significance (*p* = 0.0453; *p* = 0.0297). Generally, the combination of Startec with biostimulants (either M, Bac or Bac_M) resulted in lower average yields in each year, with significant deficits in 2016 (*p* < 0.001) and 2017 (*p* = 0.0075).

The application of Bac_M on control plots (without additional fertilization to pre-treatment with pig slurry) (Bac_M_0) resulted in higher yields in 2015 and 2016, while in 2017 the yield was lower. The differences ranging from 11.5% (1.8 t) in 2015 over 14% (3.2 t) in 2016 to 8.4% (2.2 t) in 2017, were of statistical significance in 2016 (*p* = 0.032). When Bac was applied on control plots as the only biostimulant, significant differences were present in 2017 in form of a lower yield by 12.3% (3.2 t) (*p* = 0.0253). These, and all further average yield data of the fertilizer treatments from 2015 to 2017, are shown in Figure 3.

The corncob ratios of the four different fertilizers (Table 2) were not influenced by the application of biostimulants, but by the yield. Higher yields tended to result in lower corncob ratios, while lower yields were correlated with higher corncob ratios. Statistical differences were predominantly present in 2015, not in 2016 and only in three cases in 2017.

## 4. Discussion

### 4.1. Effect of Mycorrhiza Application

Unexpectedly, applying mycorrhiza was found to have a negative effect on the maize yield in this study. Koide (1984) [39] observed a negative impact of mycorrhiza on sunflowers, if the amount of plant-available phosphorus in the soil was too small (15 mg P per kg soil). The soil present on the study site can be classified as moderately rich in phosphorus (15 mg kg^−1^), thus resembling the P content present in the soil used in the greenhouse experiment carried out by Koide in 1984. Furthermore, the phosphorus intakes of maize and sunflower are comparable, which might result in similar interactions between mycorrhiza, soil-P and plants [40]. Although the P inputs with the fertilizer variants differed drastically in the present study (Table 1), no correlations were found between the P input and the mycorrhizal effect on plant growth. The fertilizer treatment with the highest P inputs, viz. DAP, showed the same pattern of yield losses in combination with mycorrhiza as in the case of WolfNP, which was the fertilizer variant with the lowest P influx. These findings imply that other factors, instead of the different amounts of P present on the plots, led to the negative mycorrhiza effect that was observed to have on yields. The low influence of different nutrient inputs is also highlighted by the acceptable yields from control plots without additional fertilization (only slurry), which were statistically and economically comparable to the plots fertilized with DAP in each year and WolfNP and Startec in two out of three years, albeit the latter two comparisons would be of economic concern. However, further studies define a lower nutrient supply as a precondition for beneficial interactions between mycorrhizal fungi and plants [4,26,27]. The study site represents a typical well-supplied agricultural soil in a central European hotspot for high densities of livestock units. Furthermore, bioavailability for nutrients, especially phosphorus, should not be limited by the strong formation of stable inorganic forms, due to low levels of divalent cations present in a sandy non-acidic soil. Thus, the nutrient deficiency required for symbiosis-related benefits described by other authors for maize may not be achieved.

In 2015 and, to a certain extent, 2016, the low precipitation was definitely yield-limiting, which suppresses carbon acquisition for plant growth and plants’ supply of the mycobiont. This hypothesis is supported by the fact that the negative difference in yield gained with DAP and WolfNP, in combination with mycorrhiza observed in 2015 and 2016, is lower or not present in the wet year of 2017, when water deficiency did not inhibit carbon acquisition. When mycorrhiza was applied singly, this resulted in lower yields compared to all fertilizer variants (collective yield) without mycorrhiza, or mycorrhiza in combination with soil bacteria, in 2015 and 2016. The year of 2015 was especially dry in spring, but the total precipitation during the vegetation period was 372 mm, and thus considerably higher than the 278 mm precipitated in the 2016 vegetation period. Without doubt, these conditions will influence the process in the mycorrhizosphere. Osmotic conditions are also reported to have a strong impact on mycorrhizal colonization. Well-watered plants can have higher mycorrhizal colonization [41]. During the wet year of 2017 (473 mm rainfall in the vegetation period), the impacts of mycorrhizal application on the different fertilizer variants were not significant in general, and thus less negative than those reported in 2015 and, to a certain extent, also 2016. In other words, the above mentioned limitation of water for carbon acquisition of the plant, which also supplies the mycobiont, was not present in 2017. Maybe the mycorrhizal colonization was higher in that year, and advantages in nutrient uptake to a certain extend occurred by a symbiotic relationship between plant and fungi.

Furthermore, we speculate that the negative impact may be caused by effects of microbial communities naturally present in the rhizosphere on the study site. This could also contribute to explaining the role of Startec in our study. In combination with Startec, the negative impact of mycorrhiza is highest (about 15% yield reduction over three years). The combination of Startec with biostimulants, independently of the type, resulted in lower yields in each year. These deficits reached a significant extent in 2016 and 2017. In 2015, the loss of 6.8% due to the application of biostimulants was not statistically significant on Startec plots, but would be economically relevant in practice. Startec’s organic compounds, such as oil cake and bonemeal, have probably increased microbial activity of the naturally present microbiome of the study site, which may not be positively influenced by mycorrhiza or soil bacteria application, or the other way around. Oil cakes are often used to raise the microbial metabolism in the bioremediation of soils [42] and other types of biotechnological application [43]. Bonemeal is known to increase the mineralization dynamics in soils [44], and act as a biostimulant for bacteria [45]. It is possible that the organic compounds in Startec stimulated the mycorrhiza-relevant soil microorganisms on the study site in a manner that provoked parasitic actions by the applied mycorrhiza, or affected the mycorrhiza itself independently from other soil microorganisms. Microbial constellations are known to influence the forms of relationships between mycorrhiza and plants [16,19,20,21,22,23,24,25]. It is also conceivable that Startec’s organic compounds stimulated naturally occurring mycorrhizal helper bacteria, causing the over-colonization of mycorrhiza [46].

The average yields during the three years period of the other two fertilizers, DAP and WolfNP, did not profit from any combination with biostimulants (neither Bac or M nor Bac_M). The corncob ratios were adversely influenced by the yield, not by the application of biostimulants. Due to mainly insignificant differences of the corncob ratio, the implementation of that qualitative factor in an agroeconomic calculation for biogas production under consideration of the yield data resulted in the consequence that all combinations with biostimulants were economically inefficient. One exception from that rule occurred in the case of the conjoint application of bacteria and mycorrhiza without additional fertilizer (Bac_M_0) on the control plots. Bac_M_0 showed a higher average yield (and biogas yield) over the three year period than DAP, and was equal to WolfNP. Furthermore, the yield losses of the control variant, if combined with mycorrhiza or bacteria, were smaller than the ones on the plots where fertilizer were combined with Bac or M in comparison to the fertilizer applied without biostimulants. This trend may support the hypothesis of unfavorable interactions between the introduced bacteria and mycorrhiza and the soil microbiome of the study site, which is probably influenced more on sites with additional fertilization.

### 4.2. Interaction between Mycorrhiza and the Application of Bacillus velezensis

While the single application of soil bacteria did not result in significant changes compared to the collective yield gained with the other fertilizing variants without biostimulants and with mycorrhiza alone, the combination of soil bacteria and mycorrhiza showed some interesting interactions. Compared to the single application of mycorrhiza, the combination of the two biostimulants raised the yield over all fertilizer variants to a significant extent in 2015 and 2016. In the wet year of 2017, no differences of statistical relevance occurred. Over the different years, varying effects of inoculated soil bacteria and mycorrhiza are also reported by Nacoon et al., 2021 [47]. The observed interactions between the biostimulants and their effects on the fertilizer variants may be explained by changes in the microbial communities due to introduced soil bacteria, forming a milieu in the soil that alleviated the assumed parasitic interactions of the mycorrhiza. The effects of applying soil bacteria are described in the literature as stimulating the microbial communities in agricultural soils to support plant nutrient uptake through the biochemical mobilization of phosphorus, to produce plant-relevant signal molecules or to act as antagonists to soil-borne plant diseases provoked by *Fusarium* spp. [48,49,50,51,52,53,54,55,56]. Changes in the soil microbiome through soil bacteria treatments can be stable for a period of month [57,58,59] and highly specific in their effects on mycorrhiza [19,21,59], and may thus play a crucial role in the present study. In a greenhouse experiment, Yadav et al., 2020 [20] reported that naturally occurring rhizobacteria were more effective in promoting nutrient uptake and growth in mycorrhiza-treated wheat plants than non-native bacteria. Jäderlund et al., 2008 [19] showed that mycorrhiza can have a positive effect or no effect on plants’ resistance to a pathogenic fungus growth, depending on the soil bacteria species inoculated. The same bacteria that neutralized the positive mycorrhizal influence on wheat in the experiment by Jäderlund have also been reported to have a beneficial effect on clover [59]. These results make it conceivable that the microbial constellations on the study site provoked adverse mycorrhizal effects, which can be influenced by the application of *Bacillus velezensis*.

Our results point to the importance of the analysis of the soil microbiome and the simultaneous consideration of the soil and its nutrients, as well as the factor of the crop to be promoted, as precondition of the successful application and further development of mycorrhiza and comparable products in agricultural practice. The basis of a robust analysis is a better understanding of the interrelated abiotic and biotic processes within the soil influencing the effect of these products.

## 5. Conclusions

The application of the mycorrhizal fungi *Rhizoglomus intraradices* resulted in lower yields during the two dry years in the study. It is conceivable that the sufficient soil nutrient supply, and a hindered carbon acquisition by the plants due to water deficiency, impede the beneficial effects on maize plants at the study site. Further, an over-colonization by mycorrhiza or other non-beneficial microbial interactions in the rhizosphere with naturally occurring microorganisms cannot be ruled out.

The combined application of the mycorrhiza and *Bacillus velezensis* alleviated the negative effect of the mycorrhiza in 2015 and 2016, and was reported to have positive effects on single fertilizer variants, above all on control plots only fertilized with pig slurry.

Both biostimulants resulted mostly in insignificant or negative effects on the yields of the different fertilizers, and must be considered to be ineffective in an agroeconomical view under the conditions present on the study site.

Further studies have to be carried out as a parallel setup on different soil types with varying nutrient status, including extensive analyses of the soil microbiome, to prove the effects of mycorrhiza under different abiotic and biotic conditions. For the broad acceptance of mycorrhiza and soil bacteria products, the perspectives and limitations in agricultural practice have to become better predictable.

## Figures and Tables

**Figure 1 microorganisms-11-01663-f001:**
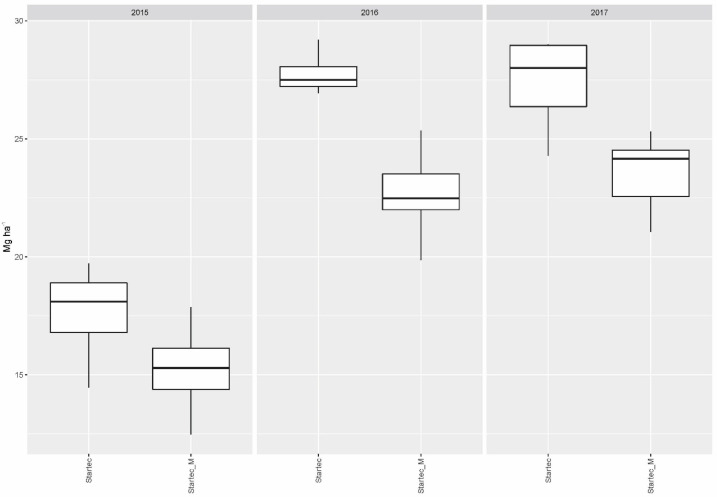
Dry matter yield gained with organomineral microgranular fertilizer applied singly (Startec) and in combination with mycorrhiza (Startec_M).

**Figure 2 microorganisms-11-01663-f002:**
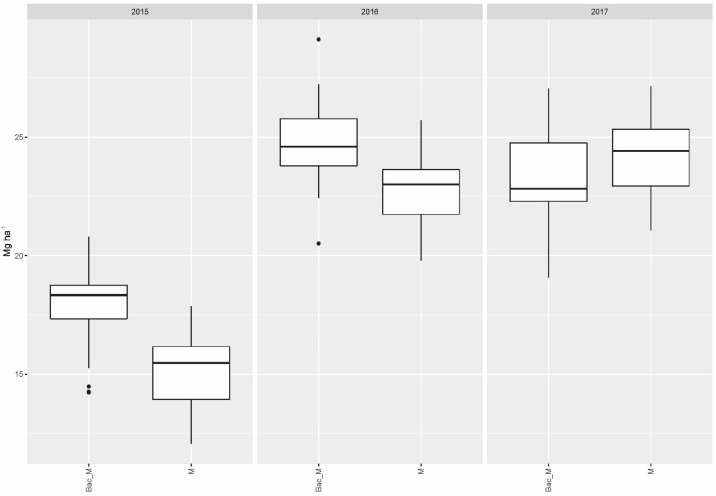
Collective dry matter yields of all fertilizer variants when applied jointly with mycorrhiza as the only biostimulant (M) and in combination with *Bacillus velezensis* and mycorrhiza (Bac_M).

**Figure 3 microorganisms-11-01663-f003:**
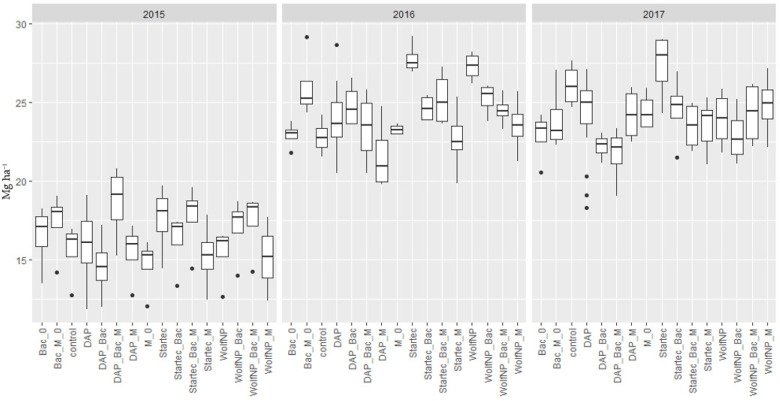
Dry matter yield of all fertilizer variants (DAP, WolfNP and Startec) from 2015 to 2017. A 0 represents combinations without additional fertilization to the pretreatment with pig slurry. Bac represents applications of *Bacillus velezensis* with and without (0) additional fertilizer. M shows data from plots where mycorrhiza were applied with and without (0) additional fertilizer. Bac_M denominate plots where mycorrhiza and *Bacillus velezensis* were applied jointly with and without (0) additional fertilizer.

**Table 1 microorganisms-11-01663-t001:** Application rates and nutrient inputs per hectare of fertilizer used.

		Nutrient Type and Input Rate Per Hectare							
Fertilizer (Type ^1^)	Application Rate Per Hectare in kg	N	P	K	S	Mg	Fe	Mn	Zn
Diammonium Phosphate (mineral fertilizer)	100	18	20	-	-	-	-	-	-
Wolf-nutraxP^®^ (mineral microgranular fertilizer)	100	11	0.2	-	8.8	7.5	-	0.4	0.1
Startec	25	1.75	2.4	0.8	1	-	0.125 ^2^	0.125 ^2^	0.375 ^2,3^
(organomineral microgranular fertilizer)									
Pre-treatment of the soil with pig slurry	20,000	172	39	98	48	21.6	-	-	-

^1^ For seed band application. ^2^ EDTA-chelated. ^3^ EDTA-chelated and as oxide.-Data missing or no data available.

**Table 2 microorganisms-11-01663-t002:** Average corncob ratio in % of fertilizer variants pig slurry (Null), diammonium phosphate (DAP), mineral microgranular fertilizer (WolfNP), organomineral microgranular fertilizer (Startec) and their combinations with *Bacillus velezensis* (Bac), Mycorrhiza (M) or both (Bac_M) from 2015 to 2017.

Fertilizer Combination	Only Slurry (Null/Control)	Null Bac	Null M	Null Bac_M	DAP	DAP Bac	DAP M	DAP Bac_M
Year 2015	59.3	57.0	63.5	59.3	60.4	60.5	62.3	54.0
Year 2016	67.0	67.5	66.8	68.0	67.5	66.8	64.5	68.0
Year 2017	55.8	59.0	57.5	57.8	57.4	61.5	57.5	61.0
**Fertilizer** **Combination**	**WolfNP**	**WolfNP Bac**	**WolfNP M**	**WolfNP Bac_M**	**Startec**	**Startec Bac**	**Startec M**	**Startec Bac_M**
Year 2015	64.0	59.8	63.0	55.3	62.0	58.5	65.3	58.0
Year 2016	66.5	65.8	65.5	67.5	67.3	68.0	67.4	67.8
Year 2017	58.3	59.3	57.0	57.3	56.3	55.3	58.9	59.3

## Data Availability

The datasets generated during and/or analyzed during the current study are available from the corresponding author upon reasonable request.

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
