# Peer review of "On the Negative Impact of Mycorrhiza Application on Maize Plants (Zea mays) Amended with Mineral and Organic Fertilizer"

_microorganisms, 2023, doi:10.3390/microorganisms11071663_

Round 1
Reviewer 1 Report
This is an interesting manuscript about the effect of mycorrhiza and soil bacteria application on fertile soil as a field experiment over three years near Wadersloh, western Germany with a European Atlantic climate. The study applied standard fertilizer (diammonium phosphate) and two micro-granular fertilizers (mineral and organomineral) alone or in combination with the biostimulants mycorrhiza and/or Bacillus velezensis as a soil bacteria.
The present work was organized logically, and the results obtained were reliable and persuasive. The results are well presented, their interpretation is relevant, and the methods are highly detailed. I would therefore recommend accepting this manuscript as a limitedly focused paper.
Author Response
Dear Reviewer,
thank you for reading the manuscript and for recommending the acceptance.
During the last weeks we used the input of several reviewers and an academic editor to get the manuscript into its present form. That may also have been a reason why you have been ok with our work.
Thank you and the other reviewer for your support!
Best wishes
Matthias Thielicke
Reviewer 2 Report
The manuscript of Thielicke et al. is devoted to the description of the negative impact of the introduction of fungal spores on the yield of maize. I agree with the authors that the publication of negative results is important for the development of science. But at the same time, authors must present their results in accordance with scientific standards. I have serious doubts about the scientific value of this manuscript in the format in which it was presented.
Basic remarks:
1) The materials used to cultivate the soil and plants are described very poorly. Because of this, it is difficult to understand the agrosystem that the authors analyze. Lines 74-80 require a much more detailed description (see minor remarks). Line 124 does not answer what the authors used as mycorrhiza. - But this is the main topic of the manuscripte.
2) Mycorrhiza is an associative symbiosis between plants and mycorrhizal fungi. After reading the manuscript, I realized that the authors mean something else by this term, but what exactly is not explained in the text. For example, one of the key words is "endomycorrhiza", but no experimental evidence of the occurrence of symbiosis between "mycorrhiza" and corn is given in the described experiments. (From the null hypothesis) I see no reason to believe that mycorrhiza occurred in the described experiments.
3) In the introduction (lines 53-55) the authors mention the specificity of mycorrhizal symbioses. At the same time, no information is provided on the specificity of mycorrhizal fungi and maize used in the experiment. On what basis did the authors expect to obtain an increase in yield on maize with the application of the biostimulants used. Are there any laboratory or greenhouse experiments that have demonstrated the promise of using them as biostimulants in relation to maize? Overall, it seems that the authors took a random culture of mycorrhizal fungi and experimented with maize plants. And they do not even report what kind of mycorrhizal fungi were used. I cannot call such an approach scientific.
Minor remarks:
1) Add to the Introduction section a discussion of research and application of maize mycorrhiza.
2) Lines 74-80 – indicate in more detail the name of the fertilizers used in the work and their manufacturers.
3) Lines 81-82 - were the plots the same for the variants during the 3 years of the experiment?
4) Line 101 - indicate the source of seeds of corn cultivar Farmpilot.
5) Line 124 - What is mycorrhiza here? And how was it grown?
6) Lines 194-198 - table 2 contains information in full presented in figure 3? If yes, then table 2 should be deleted.
7) Figure 3 – How do the authors explain the inefficiency or low efficiency of any stimulants (including mineral and organic) in their experiment? Especially in 2017?
8) Lines 214-220 - how close are the phosphorus intakes of maize and sunflower?
9) Line 330 - "the sufficient soil nutrient supply" is written, but lines 214-220 discuss that there was not enough phosphorus in the soil for mycorrhiza to occur. Which of the statements is true?
Author Response
Dear Reviewer,
please find our answer to your work attached.
Best wishes
Matthias Thielicke

Round 2
Reviewer 2 Report
I thank the authors for the detailed response to my remarks. I am completely satisfied with their answer. However, since the publication of the manuscript is planned in the journal "Microorganisms", the authors need to further clarify the names and characteristics of the microorganisms used in the work.
Minor remarks:
1. Line 133 – Add a description of the used Rhizoglomus intraradices: source of isolation, source of obtaining, name of the strain or number in any collection, or any other information characterizing the culture of mycorrhizal fungus used.
2. Line 140 – on the ABiTEP GmbH website it says " Wir entwickeln und produzieren biologische Biostimulanzien und Pflanzenschutzmittel, unter anderem auf Basis der Bakterienstämme Bacillus velezensis FZB42 und Bacillus atrophaeus ABi 05." (https://abitep.de/). If the authors used a biological product with the Bacillus velezensis FZB42, the strain number (FZB42) should be indicated in the Materials and Methods section.
3. Authors should use one abbreviation for the word "ton". In lines 110, 181 and in figures 1-3, "Mg" is used; and in lines 186-187, 192-195, "t" is used. Since the authors use Mg as magnesium in Table 1, tons should be denoted by "t".
Author Response
Dear Reviewer,
Thank you again for the further remarks, resulting from your attentive reading of the manuscript.
We added the information characterizing the culture of our Rhizoglomus intraradices. Also the strain number and the custodian, where the reader can obtain cell cultures or the sequence of Bacillus velezensis (FZB42) is complemented in the material and methods part.
The unconstant usage of tons and mega Gramm has been changed into “t” in the text.
The latest changes according to your comments on the material and methods are highlighted in light green.
In the name of all authors I want to express our gratefulness for the supervision from the first to the present version of the manuscript.
Best wishes
Matthias Thielicke